# Factors Affecting the Simultaneous Removal of Nitrate and Reactive Black 5 Dye via Hydrogen-Based Denitrification

**Tippawan Singhopon [1], Kenta Shinoda [1], Suphatchai Rujakom [2] and Futaba Kazama [2,*]**

[1] Graduate School of Medicine, Engineering, and Agricultural Sciences, University of Yamanashi, 4-4-37 Takeda, Kofu, Yamanashi 400-8510, Japan; Ptippawan.sing@gmail.com (T.S.); kentamthshinoda@gmail.com (K.S.)

[2] Interdisciplinary Research Centre for River Basin Environment, University of Yamanashi, 4-4-37 Takeda, Kofu, Yamanashi 400-8510, Japan; suphatchai.r@gmail.com

[*] Correspondence: kfutaba@yamanashi.ac.jp

**Abstract:** Textile wastewater (TW) contains toxic pollutants that pose both environmental and human health risks. Reportedly, some of these pollutants, including $NO_3^-$, $NO_2^-$ and reactive black 5 (RB-5) dye, can be removed via hydrogen-based denitrification (HD); however, it is still unclear how different factors affect their simultaneous removal. This study aimed to investigate the effect of $H_2$ flow rate, the sparging cycle of air and $H_2$, and initial dye concentration on the TW treatment process. Thus, two reactors, an anaerobic HD reactor and a combined aerobic/anaerobic HD reactor, were used to investigate the treatment performance. The results obtained that increasing the $H_2$ flow rate in the anaerobic HD reactor increased nitrogen removal and decolorization removal rates. Further, increasing the time for anaerobic treatment significantly enhanced the pollutant removal rate in the combined reactor. Furthermore, an increase in initial dye concentration resulted in lower nitrogen removal rates. Additionally, some of the dye was decolorized during the HD process via bacterial degradation, and increasing the initial dye concentration resulted in a decrease in the decolorization rate. Bacterial communities, including Xanthomonadaceae, Rhodocyclaceae, and *Thauera* spp., are presented as the microbial species that play a key role in the mechanisms related to nitrogen removal and RB-5 decolorization under both HD conditions. However, both reactors showed similar treatment efficiencies; hence, based on these results, the use of a combined aerobic/anaerobic HD system should be used to reduce organic/inorganic pollutant contents in real textile wastewater before discharging is recommended.

**Keywords:** biodegradation; decolorization removal efficiency; hydrogen-based denitrification; microbial community; nitrate removal efficiency; synthetic textile wastewater

## 1. Introduction

Globally, textile industry wastewater, which contains high concentrations of organic and inorganic pollutants, is a serious environmental issue, especially in developing countries, such as Thailand, India, and China. Reportedly, approximately 17–20% of the pollutants in textile wastewater are typically released during the dye manufacturing and textile finishing processes, which involve the use of large amounts of water [1]. Generally, textile industry wastewater contains approximately 72 toxic compounds, of which approximately 30 cannot be removed via the application of traditional wastewater treatment processes [2]. Therefore, even after treatment using traditional treatment processes, textile industry effluent still contains harmful compounds (predominantly originate from anthropogenic sources) discharged into nature sources. The presence of these contaminants in water decreases dissolved oxygen content, increases insoluble materials, and prevents sunlight penetration [3,4].

The compounds used in the dyeing process, which are the most harmful, are considered to be the major pollutants in textile wastewater [5]. Annually, dye consumption is estimated at more than 10,000 tons, and reportedly, approximately 10% of textile wastewater, containing mutagens and carcinogens that negatively affect human health, is discharged into the natural environment without treatment [6]. A previous study revealed that even at low concentrations (approximately 1 mg/L), dyes appear visible in water; however, the actual concentration of dyes in textile wastewater falls within the range ~10–200 mg/L [7]. Reportedly, the removal of dyes in wastewater can be realized using physical-chemical-biological approaches, such as membrane filtration, flocculation-coagulation, ion exchange, and advanced oxidation processes, which have been developed in several studies [8]. Generally, biological treatment methods (e.g., anaerobic-aerobic degradation) are preferred given their environmentally friendly nature, their low operation cost, and the fact that they offer the possibility to realize the complete mineralization of these pollutants [9]. The mechanism of dye degradation proceeds as follows: first, the azo bond (-N=N-) is cleaved by azo reductase via redox processes under anaerobic conditions. Thus, the dyes are decolorized, and amino groups (-NH$_2$) are formed alongside several intermediates, such as amines. Second, -NH$_2$ and other intermediate metabolites are easily degraded under aerobic conditions, as shown in the proposed mechanism by which azo dyes are reduced by bacterial cells (Figure 1) [10]. Therefore, to improve the treatment of textile wastewater before discharging, most textile industries employ the combined aerobic-anaerobic process due to high removal efficiency, low-cost, simple to maintain and suitability for removing organic/inorganic carbon and dye color [11].

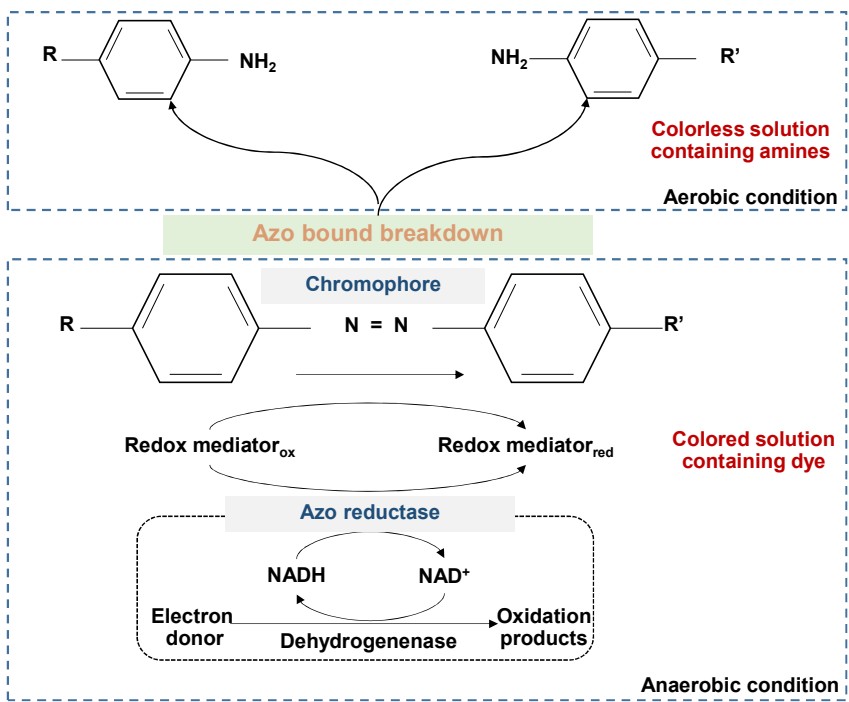

**Figure 1.** Proposed mechanism for reduction of dye by bacterial cell adaptation from [12].

The nitrate ion (NO$_3^-$), which can be converted to the nitrite ion (NO$_2^-$), is a common contaminant that is usually detected in groundwater, surface water, and wastewater [13]. Usually, its concentration in textile wastewater ranges between 150–700 mg-N/L given that in the textile industry, it is generally used to improve the quality of fibers in dyeing baths. However, at such concentrations, it is rather harmful to aquatic organisms and poses a serious risk to human health (the guideline values provided by WHO for NO$_3^-$ and NO$_2^-$ are 11.3 and 0.91 mg-N/L) [14]. Therefore, based on desirable removal rates, cost,

and simplicity, several treatment processes, including ion exchange, chemical denitrification, and reverse osmosis, have been employed in the removal of these impurities from wastewater [15]. However, in this regard, biological denitrification is widely practiced considering that it has several benefits. Specifically, it is environmentally friendly, low-cost, and results in the generation of fewer byproducts that do not require posttreatment [16]. Additionally, there are two types of denitrification processes, namely, heterotrophic and autotrophic denitrification. Heterotrophic denitrification requires organic carbon compounds, such as ethanol, methanol, or acetate, which function as a carbon source and an electron donor. In contrast, autotrophic denitrification utilizes hydrogen ($H_2$), iron (Fe), or sulfur (S) as the electron donor and carbon dioxide ($CO_2$) or bicarbonate ($HCO_3^-$) as the inorganic carbon source [17]. Reportedly, the denitrification rate associated with autotrophic denitrification is higher than that associated with heterotrophic denitrification [18]. In hydrogen-based autotrophic denitrification (HD), $H_2$ gas is used as an electron donor to remove $NO_3^-$ and $NO_2^-$ from wastewater. The advantages of this process include: (1) the low cost of $H_2$ as an electron donor compared with the use of carbon compounds, such as methanol or acetate, (2) low sludge accumulation, and (3) the low solubility of $H_2$ [19]. The theoretical equation for HD using bicarbonate as an inorganic carbon source and $H_2$ as an electron donor is, as shown in Equation (1) [20].

$$NO_3^- + 2.892H_2 + 0.171HCO_3^- \rightarrow 0.483N_2 + 2.268H_2O + 0.17OH^- + 0.034C_5H_7NO_2 \qquad (1)$$

Additionally, previous studies attempted to develop the HD process and evaluated its performance in relation to the simultaneous treatment of various pollutants in textile wastewater, especially $NO_3^-$, $NO_2^-$ and dye [21]. Their results showed the possibility of the simultaneous reduction of nitrogen and a small amount of dye in both anaerobic and combined aerobic/anaerobic reactors. In an anaerobic reactor that was operated by continuous feeding with $H_2$ gas, they observed the complete removal of $NO_3^-$ and $NO_2^-$ as well as a decrease in the color intensity of the RB-5 dye. Similarly, a combined aerobic/anaerobic reactor that is commonly used to reduce organic/inorganic pollutant contents in textile wastewater was set-up by feeding with air and $H_2$ gas to evaluate the treatment efficiency of the HD system coupled with aerobic conditions mainly to find out other ways to reduce the cost of $H_2$ gas supply. In real wastewater treatment plants, most aerobic bacteria play a significant role in the removal of organic pollutants as well as some toxic chemicals from wastewater.

Therefore, in this study, the anaerobic and a combined HD system was continuously operated that aimed to identify (1) the key factors affecting nitrogen removal efficiency (NRE) and decolorization removal efficiency (DRE) were determined by employing different conditions of $H_2$ flow rate, the sparging cycle of air and $H_2$ supply, and initial dye concentration. Further, (2) the major bacterial species involved in the system were identified so as to clarify the mechanisms involving bacterial activity and pollutant removal.

## 2. Materials and Methods

### 2.1. Synthetic Textile Wastewater

Synthetic textile wastewater was prepared based on the quality of textile wastewater in Thailand, which has a composition as previously described: 486 mg $NaNO_3$/L (80 mg $NO_3$–N/L), 110 mg $KH_2PO_4$/L, 1000 mg $NaHCO_3$/L, 21 mg $CaCl_2$/L, 38 mg $MgSO_4 \cdot 7H_2O$/L, and 1.0 mL of trace elements (I) and (II) [21,22]. Trace element I contained 50 mg/L EDTA and 50 mg/L $FeSO_4$, whereas trace element II contained 150 mg/L EDTA, 430 mg/L $ZnSO_4 \cdot 7H_2O$, 240 mg/L $CoCl_2 \cdot 6H_2O$, 10 mg/L $MnCl_2 \cdot 4H_2O$, 250 mg/L $CuSO_4 \cdot 6H_2O$, 220 mg/L $MnCl_2 \cdot 5H_2O$, 220 mg/L $NaMoO_4 \cdot 5H_2O$, 190 mg/L $NiCl_2 \cdot 6H_2O$, 210 mg/L $NaSeO_4 \cdot 10H_2O$, and 14 mg/L $H_3BO_4$. Further, the influent $NO_3$–N concentration was fixed at 80 mg-N/L for all the experiments, and reactive black 5 (RB-5) or Remazol black B, which is often used in the textile industry and was obtained from Sigma-Aldrich (USA),

was selected as the azo dye for this study. It has a diazo-vinylsulfonic structure ($C_{26}H_{21}N_5Na_4O_{19}S_6$) with a molecular weight of 991.82 g/mol that can be detected at a maximum wavelength ($\lambda_{max}$) of 597 nm [23]. Furthermore, in this study, the concentrations of the azo dye were varied (10, 15, 30, and 60 mg/L) depending on the operating conditions of each phase, as described in Table 1.

### 2.2. Experimental Setup and Operation Conditions

Two reactors from a previous study that had been enriched for more than 60 days were used in this study [22]. The reactors, both with a working volume of 2 L, were fabricated from a plastic beaker and were operated at 30–35 °C. A schematic diagram of the experimental setup is shown in Figure 2, and the operating conditions employed are presented in Table 1.

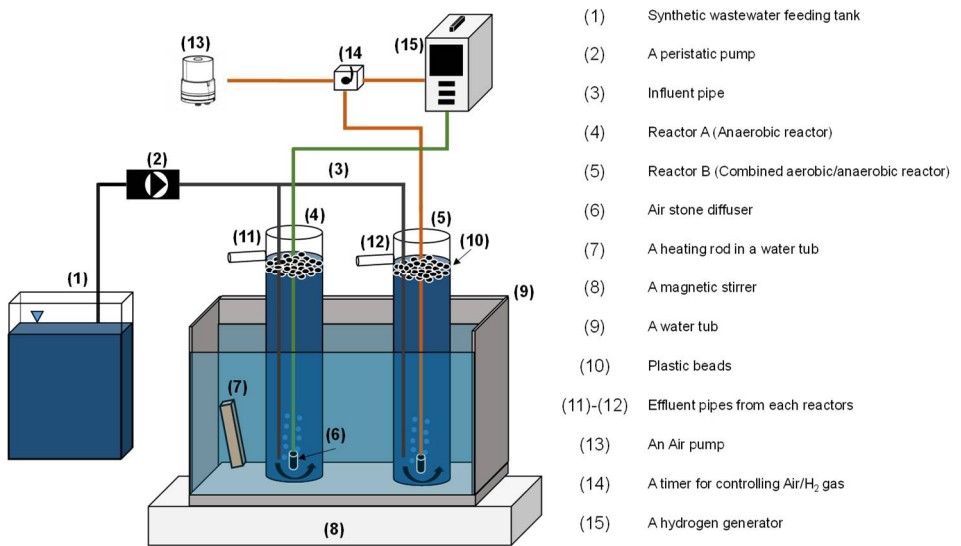

| | |
|---|---|
| (1) | Synthetic wastewater feeding tank |
| (2) | A peristatic pump |
| (3) | Influent pipe |
| (4) | Reactor A (Anaerobic reactor) |
| (5) | Reactor B (Combined aerobic/anaerobic reactor) |
| (6) | Air stone diffuser |
| (7) | A heating rod in a water tub |
| (8) | A magnetic stirrer |
| (9) | A water tub |
| (10) | Plastic beads |
| (11)-(12) | Effluent pipes from each reactors |
| (13) | An Air pump |
| (14) | A timer for controlling Air/$H_2$ gas |
| (15) | A hydrogen generator |

**Figure 2.** Schematic diagram of the experimental setup.

**Table 1.** Summary of operation conditions.

| Run | Influent Concentrations | | Reactor A (Anaerobic) | Reactor B (Combined Aerobic /Anaerobic [1]) | Operation Time (day) |
|---|---|---|---|---|---|
| | NO$_3$–N (mg-N/L) | RB-5 Dye (mg/L) | H$_2$ Gas (mL/min) | Air/H$_2$ Gas (min) | |
| 1 | 80 | 10 | 100 | 30/60 | 1–20 |
| 2 | 80 | 10 | 50 | 10/60 | 21–40 |
| 3 | 80 | 15 | 50 | 10/60 | 41–60 |
| 4 | 80 | 30 | 50 | 10/60 | 61–80 |
| 5 | 80 | 60 | 50 | 10/60 | 81–100 |

To observe the performances of reactors A (anaerobic) and B (aerobic/anaerobic) in terms of NRE and DRE runs 1 and 2 were operated with several H$_2$ flow rates and sparging cycles of air and H$_2$, respectively, for 40 days. For reactor A, the flow rate of H$_2$ gas was varied from 50 to 100 mL/min, and for reactor B, the sparging cycle of air and H$_2$ was varied from 30 min–air/60 min-H$_2$ to10 min–air/60 min-H$_2$. Further, to control the aerobic and anaerobic conditions using a timer, the sparging rates of air and H$_2$ in reactor B were fixed at 50 mL/min. H$_2$ gas was supplied from a H$_2$ generator, while the air was supplied from an air pump via an air stone diffuser. The influent concentration was maintained at 80 mg NO$_3$–N/L and 10 mg/L RB-5 for both reactors. Runs 3 to 5 were subsequently oper-

ated from days 41 to 100 to investigate the effect of initial RB-5 concentrations on the treatment efficiency of the systems under the desired operating conditions described in Table 1. The experiments corresponding to runs 3 to 5 were operated under different initial RB-5 concentrations, i.e., 15, 30, and 60 mg/L, with the same influent $NO_3$–N concentrations. The $H_2$ sparging rate was fixed at 50 mL/min for reactor A, while a sparging cycle of 10 min–air/60 min-$H_2$ was adopted for reactor B.

All the reactors in this experiment were subjected to multiple operation cycles with sequencing stages of decantation, filling, and reaction. The decantation and filling stages were always completed within 1 h, while effluent samples were collected at 24 h intervals, indicating that the reaction stage lasted 23 h.

### 2.3. Sampling and Analytical Methods

The collected influent and effluent samples were centrifuged for 10 min at 10,000 rpm (Hitachi CF 16RXII, Japan). Thereafter, the clear supernatant was preserved in a freezer at −18 °C for further analyses of $NO_3$–N, $NO_2$–N, and RB-5 concentrations. Dissolved oxygen (DO), pH, dissolved hydrogen (DH), and temperature measurements were carried out inside the reactors before sampling using a DO probe (YSI 58 dissolved oxygen meter, Japan), pH meter (Horiba-B712, Japan), DH meter (ENH-2000, Taiwan), and a digital thermometer (WT-6, China), respectively. $NO_3$–N, $NO_2$–N and RB-5 concentrations were colorimetrically analyzed using a spectrophotometer (UV-1800, Shimadzu-Spectrophotometer, Japan). Specifically, $NO_3$–N and $NO_2$–N concentrations were determined as previously described by APHA [24], while the concentration of RB-5 was determined using a standard curve, of which several known concentrations of RB-5 were plotted against their detected absorbance at $\lambda_{max}$ 597 nm [25]. Thereafter, the DRE of the RB-5 dye was calculated based on the Lambert–Beer law [26]. The linear equation used for RB-5 determination in this study was $y = 0.0214x + 0.0201$ ($R^2 = 0.9980$).

### 2.4. Bacterial Community Analysis

The sludge samples for bacterial community analysis were collected from both reactors after several days of operation. Sludge samples, with a wet weight of approximately 0.12 g, were collected for total DNA extraction using a FastDNA® SPIN Kit for soil analysis (MP-Biomedicals, USA). DNA concentration was determined by nanodrop using a Quantus™ fluorometer (Promega Corp., USA), and DNA samples were used for next-generation sequencing (NGS) analysis that was carried out by a commercial service in Japan (FASMAC Co., Ltd.). Sequencing was performed based on the V4 region of bacterial 16S rRNA amplified by Univ-515 F (5-GTG YCA GCM GCC GCG GTA A-3) and Univ-806R (5-GGA CTA CNV GGG TWT CTA AT-3) primers. The amplified metagenomic sequences were obtained using the MiSeq platform. The raw sequence data were then taxonomically classified using QIIME software version 1.9.0 to obtain the operational taxonomic units (OTUs), which were then clustered at 97% similarity and the relative bacterial abundance [27]. The bacterial abundances observed under different operation conditions were presented in a heatmap, which was created using R software version 4.0.2 with the Heatplus (version 2.30.0) and Vegan (version 2.5.6) packages.

### 2.5. Statistical Analysis and Calculation

To observe DRE differences under different operation conditions at 95% confidence interval, analysis of variance (ANOVA) and least significant difference (LSD) tests were performed. The nitrogen loading rate (NLR), nitrogen removal efficiency (NRE), decolorization removal efficiency (DRE), and the ratio of nitrate and nitrite degradation (ND) were calculated using Equations (2)–(5), where $A_0$, $A_t$, and $B_t$ represent initial $NO_3$–N, effluent $NO_3$–N, and effluent $NO_2$–N (mg N/L) concentrations, respectively. Q represents

the flow rate (L/day), and V represents the reactor volume ($m^3$). Moreover, $C_0$ and $C_t$ represent the absorbance values of the initial RB-5 concentration and that after treatment, respectively.

$$NLR \ (mg\text{-}N/m^3 \cdot day) = \frac{A_0 \times Q}{V} \tag{2}$$

$$NRE \ (\%) = \frac{A_0 - (A_t + B_t) \times 100}{A_0} \tag{3}$$

$$DRE \ (\%) = \frac{(C_0 - C_t) \times 100}{C_0} \tag{4}$$

$$ND = \frac{A_t + B_t}{A_0} \tag{5}$$

## 3. Results and Discussion

### 3.1. Effect of $H_2$ Flow Rate and Sparging Cycle of Air and $H_2$ Feeding on NRE Subsection

The concentrations of $NO_3$–N and $NO_2$–N in the effluent, as well as the NREs of reactors A and B under the different operating conditions that characterized runs 1 and 2, are shown in Figure 3. During the first stage, owing to the startup period of the system with a high $H_2$ sparging rate (100 mL/min of $H_2$) resulted in the accumulation of $NO_3$–N and $NO_2$–N in reactor A, which thereafter decreased to approximately zero after the startup period, as shown in Figure 3a. After 20 days, the $H_2$ flow rate was decreased to 50 mL/min so as to observe the NRE. The results obtained showed that $NO_3$–N and $NO_2$–N were completely removed. However, the NREs observed following run 1 (95 ± 16.2%) and run 2 (100 ± 0.5%) were not significantly different. Similarly, the NRE of reactor B, which was observed when the sparging cycle of air and $H_2$ was changed, also indicated low $NO_3$–N and $NO_2$–N concentrations, with NREs in the ranges 96 ± 10.9 and 98 ± 7.1% for RUNs 1 and 2, respectively, as shown in Figure 3b.

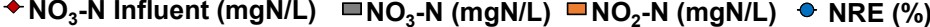

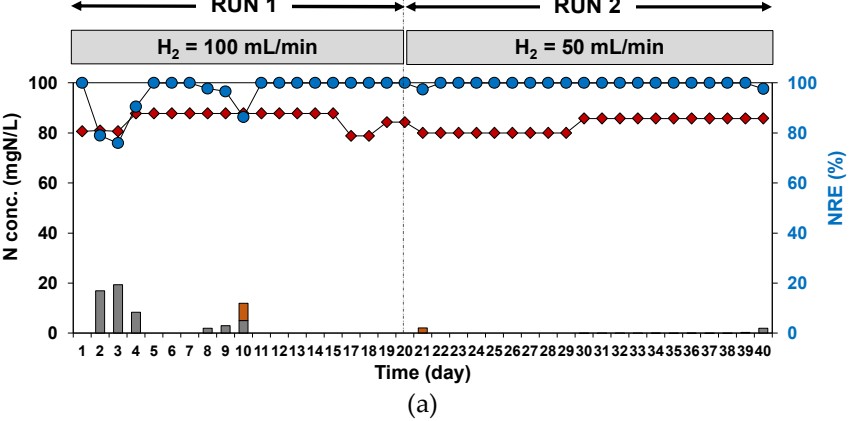

(a)

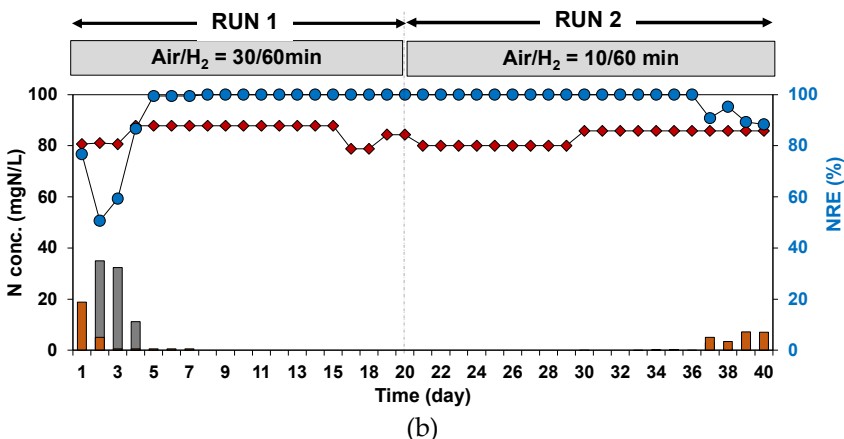

**Figure 3.** Effluent NO₃–N and NO₂–N concentrations and nitrogen removal efficiency (NRE) in (**a**) reactor A and (**b**) reactor B.

To confirm the effect of H₂ and the sparging cycle of air and H₂, a batch experiment test was conducted under a hydraulic retention time (HRT) of 48 h, and effluent samples were collected at 3 h intervals. The ratios of NO₃–N and NO₂–N degradation (ND) for reactors A and B at various periods are shown in Figure 4a,b, respectively. Under anaerobic conditions (reactor A), the ND value almost reached zero at 12 h following run 1 (100 mL/min) and 15 h following run 2 (50 mL/min), as shown in Figure 4a. This implies that run 1 provided a more favorable reduction rate. Reportedly, H₂ is the main factor associated with enhancing NRE during the HD process. This is because H₂ concentration significantly affects the process; an increase leads to a higher bacterial growth rate, which brings about a higher efficiency [28,29]. The last results reported that 0.28 moles of H₂ could reduce 1 mol of NO₃–N; however, a higher H₂ concentration should be applied to the system so as to reduce the limitation of H₂ [30,31]. Previous studies have suggested that DH should be maintained above 0.2 mg/L since NO₃–N and NO₂–N reductases can be inhibited, while NO₂–N reductase is more sensitive than NO₃–N reductase, causing NO₂–N accumulation in the system [8]. In this study, the reactors were operated at a high sparging rate of H₂ (140 and 70 L/day), i.e., higher than the required rate so as to ensure the removal of NO₃–N based on the theoretical HD equation, which shows that a sparging rate of only 1.54 L/day is required to remove 80 g-N/m³/day of NLR. In addition, the concentration of DH was found to be 1.48 ± 0.31 and 1.16 ± 0.39 mg/L in reactors A and B, respectively, indicating that the amount of DH available was sufficient for the HD process. Figure 4b shows the ND observed in reactor B when the aeration time was decreased. The results obtained following run 2 showed that the amounts of NO₃–N and NO₂–N decreased to a greater extent when the aerobic time was shorter, suggesting that a longer non-aeration time could be better for NO₃–N reduction given that oxygen usually halts the HD process. Conversely, this study showed that oxygen has no significant effect on NRE, indicating that NO₃–N and NO₂–N reductions were mostly achieved by H₂, while DO showed a partial ability to bring about NO₃–N and NO₂–N removal [32].

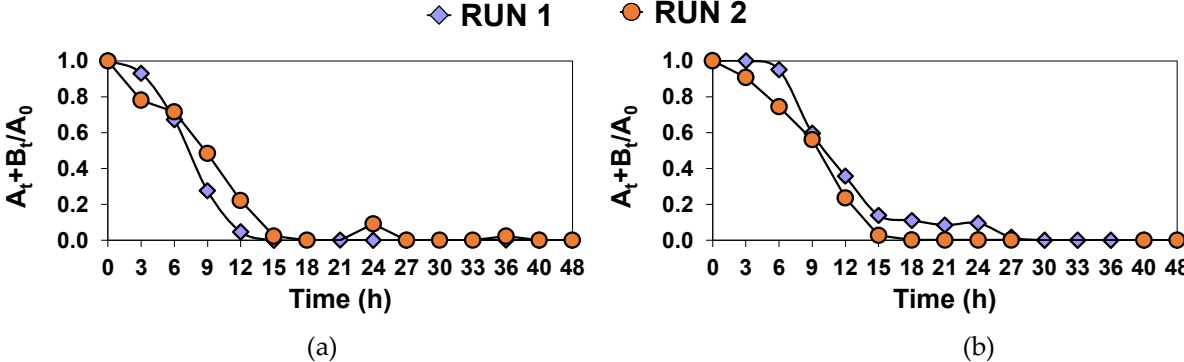

**Figure 4.** Ratio of NO₃–N and NO₂–N degradation profiles from batch testing in (**a**) reactor A and (**b**) reactor B.

Therefore, this study confirmed that the amount of $H_2$ gas and the air/$H_2$ sparging cycle are the main factors associated with improving the NRE, as seen in reactors A and B. However, RB-5 dye did not show any significant effect on the NRE of the system. Moreover, the results confirmed the NRE corresponding to both reactors were not significantly different; therefore, aerobic conditions had a small effect on NRE in the HD system.

*3.2. Effect of Initial RB-5 Dye Concentration on NRE*

The effects of different RB-5 concentrations (10, 15, 30, and 60 mg/L) on the simultaneous removal of nitrogen and RB-5 dye were investigated as the details given in Table 1 (runs 1–5), and in Figure 5, $NO_3$–N and $NO_2$–N concentrations, as well as the NREs, obtained following runs 1–5 in both reactors are presented. Reactor A, which was operated with runs 1–3 (initial RB-5 concentrations: 10 and 15 mg/L), showed great efficiency given that the observed NRE was approximately 100%, as shown in Figure 5a. Thereafter, in RUN 4, the initial RB-5 dye was increased to 30 mg/L, and unexpectedly, large amounts of $NO_3$–N and $NO_2$–N accumulated. The NRE obtained following this run was more unstable than those obtained following runs 1–3, and it was slightly lower (90 ± 9.85%), considering the high accumulations of $NO_3$–N and $NO_2$–N observed. However, when 60 mg/L of RB-5 dye was added in run 5, the effluent $NO_3$–N and $NO_2$–N concentrations were lower than those resulting from run 4. The NRE corresponding to this run was 100 ± 0.83%. The results corresponding to reactor B showed consistency with those corresponding to reactor A, as shown in Figure 5b. The NRE of reactor B was found to be in the range of 95–98% following runs 1–3, and in run 4, it dropped to 90 ± 9.23%. However, in run 5, it was approximately 100%, indicating that the addition of 60 mg/L of RB-5 dye resulted in an improvement in treatment performance. This implies that the bacteria involved in the system possibly grow well and function effectively when a higher dose of RB-5 was employed; hence, the satisfactory performance. Possibly, the high concentration of RB-5 dye, which is an electron acceptor and is toxic, hindered bacterial activity related to the removal of $NO_3$–N and $NO_2$–N, resulting in a lower NRE [33].

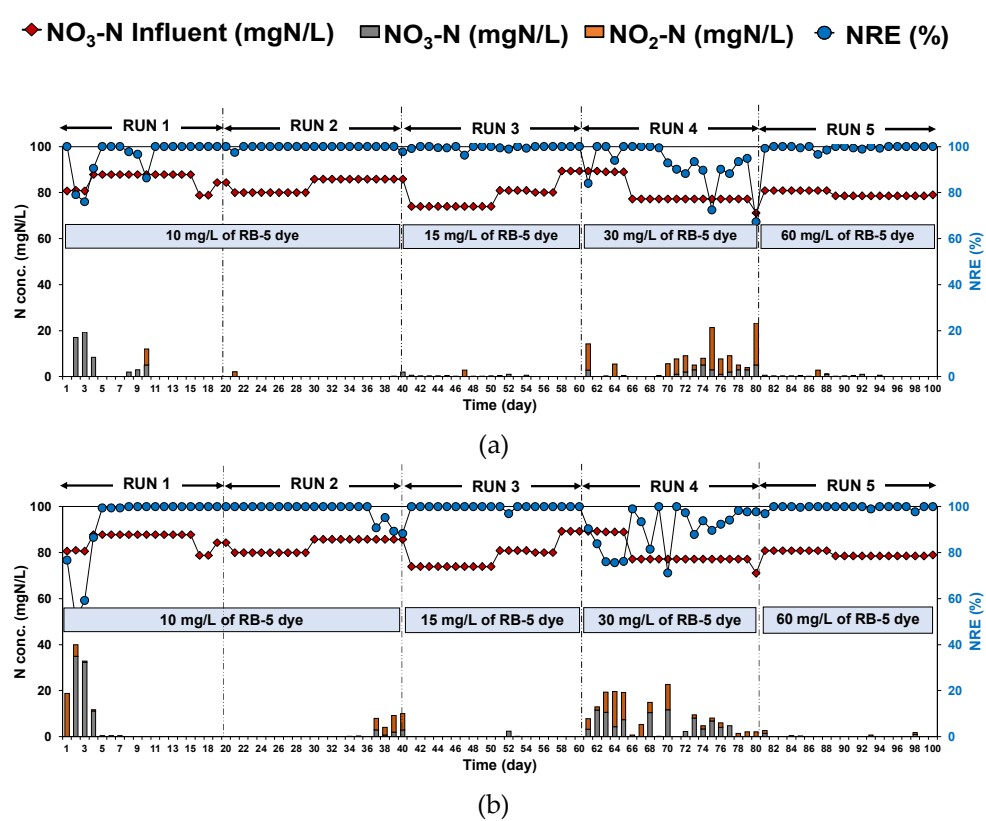

**Figure 5.** Concentrations of NO₃–N and NO₂–N in the effluent and NRE in (**a**) reactor A and (**b**) reactor B.

In conclusion, RB-5 concentration showed a slightly significant effect on both NO₃–N and NO₂–N removal. The treatment efficiency of the HD system was stable throughout the long-term operation. It was also observed that low RB-5 concentrations favored both the anaerobic and combined aerated/anaerobic systems. Conversely, a high RB-5 dose had a slight effect on NRE, which was more pronounced, especially when the initial RB-5 concentration was increased. Considering that the functional bacteria require an adaptation time in order to function properly under a high amount of toxic contamination, nitrogen removal was enhanced when the RB-5 dye was added. A previous study revealed that NO₃–N and NO₂–N could be removed when the environment is contaminated with high RB-5 concentrations, and NRE was greater when RB-5 was added to the system [21].

### 3.3. Decolorization Efficiency under Various Operating Conditions

The biodegradation of the dye can be divided into two-steps. First, the breaking of the azo bond under anaerobic conditions, and second, the degradation of amino groups as well as other intermediate metabolites under aerobic conditions [34,35]. One of the important factors that can improve DRE is the concentration of the electron donor. This is because the electron donor supports the ability of NADH to produce azo reductase, which plays a key role in the breakdown of the azo bond in the redox process [36]. Hence, in this study, the effect of the H₂ flow rate and sparging cycle of air and H₂ (as electron donors) and initial dye concentration on the performance of the HD systems was also investigated so as to observe the DRE. The average concentrations of RB-5 in the effluent and DRE under various operating conditions in runs 1–5 are shown in Table 2.

**Table 2.** The average value of effluent dye concentration and decolorization efficiency.

| Run | Initial Dye Concentration (mg/L) | Reactor A (Anaerobic) | | Reactor B (Combined Aerobic/Anaerobic) | |
|---|---|---|---|---|---|
| | | Effluent Dye Color (mg/L) | DRE (%) | Effluent Dye Color (mg/L | DRE (%) |
| 1 | 10 | 4 ± 1.5 | 55 ± 17.1 | 6 ± 1.9 | 38 ± 15.5 |
| 2 | 10 | 4 ± 1.9 | 64 ± 19.5 | 7 ± 1.7 | 45 ± 16.6 |
| 3 | 15 | 7 ± 2.9 | 59 ± 18.2 | 7 ± 2.0 | 52 ± 9.7 |
| 4 | 30 | 14 ± 4.2 | 54 ± 14.7 | 26 ± 2.8 | 20 ± 7.3 |
| 5 | 60 | 28 ± 7.5 | 56 ± 15.1 | 52 ± 8.8 | 16 ± 14.8 |

From the table, it evident that RB-5 concentrations in the effluent ranged from 4 to 7 mg/L. At an initial RB-5 concentration of 10 mg/L, the DRE was in the range of 40–65%. Additionally, no significant change in DRE was observed when a lower $H_2$ sparging rate was employed in reactor A and when a shorter aerating time was applied in reactor B. $NO_3$–N, $NO_2$–N, and RB-5 can function as electron acceptors; however, $NO_3$–N and $NO_2$–N are stronger electron acceptors than RB-5. Therefore, the denitrification process occurred during the first stage, after which RB-5 was reduced, resulting in the incomplete removal of the dye color. However, a previous study reported that the sparging rate of air and $H_2$ is a significant parameter for achieving high color removal rates in combined systems [37]. This is because the functioning of the anaerobic azo reductase enzyme, which plays a key role in the cleavage of azo bonds, is limited under aerobic conditions, resulting in an insufficient color removal rate. Initial dye concentrations also play an important role in DRE. The results obtained following runs 3–5 under various initial RB-5 concentrations, i.e., 15, 30, and 60 mg/L, indicated that increasing the initial RB-5 concentration resulted in an increase in effluent dye concentrations and a consequent decrease in DRE to values in the range 15–60%, which are lower than those obtained following runs 1–2. Previous studies have suggested that the significant parameters that can inhibit DRE depend on the toxicity and structure of the dye, inadequate biomass concentrations that can block enzymes, and initial dye concentrations [38,39]. Moreover, the results corresponding to reactor A showed that the DRE varied in the range 50–65% and was more stable than that corresponding to reactor B when a similar charging $H_2$ flow rate, air and $H_2$ sparging rate, and initial dye concentrations were employed. This observation confirmed that the bacteria in reactor A might adapt better than the bacteria in reactor B. However, at low RB-5 concentrations in the range 10–15 mg/L, both reactors A and B showed similar DRE, which decreased as the initial RB-5 concentration increased, especially in reactor B. This decrease in DRE may be attributed to the fact that the bacteria in this reactor cannot adapt to the cell as the initial dye concentration increases.

To clarify the process of RB-5 removal, Figure 6 shows the RB-5 absorbance values obtained under various HRTs (0, 12, 24, 36, and 44 h) of the initial dye concentration and the effluent following run 3. These absorbance values were determined based on the UV-vis spectra corresponding to reactors A and B. Specifically, Figure 6a,b (inset) shows the actual concentrations of RB-5 measured at 597 nm. Some reports explained that the strong absorption at $\lambda_{max}$ 597 nm represents the intensity of the chromophore group (-N=N-), which is indicative of the concentration of the dye [8,35,40]. Therefore, the reduction rate observed at different times possibly indicates the cleavage of the azo bond and the charged form resulting in the formation of other molecules. Moreover, the different peaks at 203, 210, and 246 nm could be attributed to the benzene ring, and that at 310 nm to the naphthalene ring, which are molecules formed as a result of the dye breaking process. Similarly, a previous study revealed that the new peaks at 220–230 and 260–300 nm represent the formation of sulfonate groups and aromatic amines, respectively [41]. Therefore, the difference in the spectral range between 200 and 350 nm represented the occurrence of dye degradation. For reactors A and B, the absorbance value at 597 nm under

various HRTs decreased continuously, confirming the commencement of RB-5 degradation. Thereafter, at an HRT of 24 h, a new peak clearly appeared at 263 nm in the spectra corresponding to reactor A, whereas at an HRT of 44 h, a new peak appeared at 287 nm in the spectra corresponding to reactor B, indicating the cleavage of the azo bond and the transformation of the dye into the ($-NH_2$) group. Therefore, this study showed that RB-5 dye could be decomposed in these HD systems. Further, the rate at which decolorization occurred was faster in reactor A than in reactor B due to the absence of an aerobic zone, and it dropped at a faster rate after denitrification was completed (at HRT = 36–44).

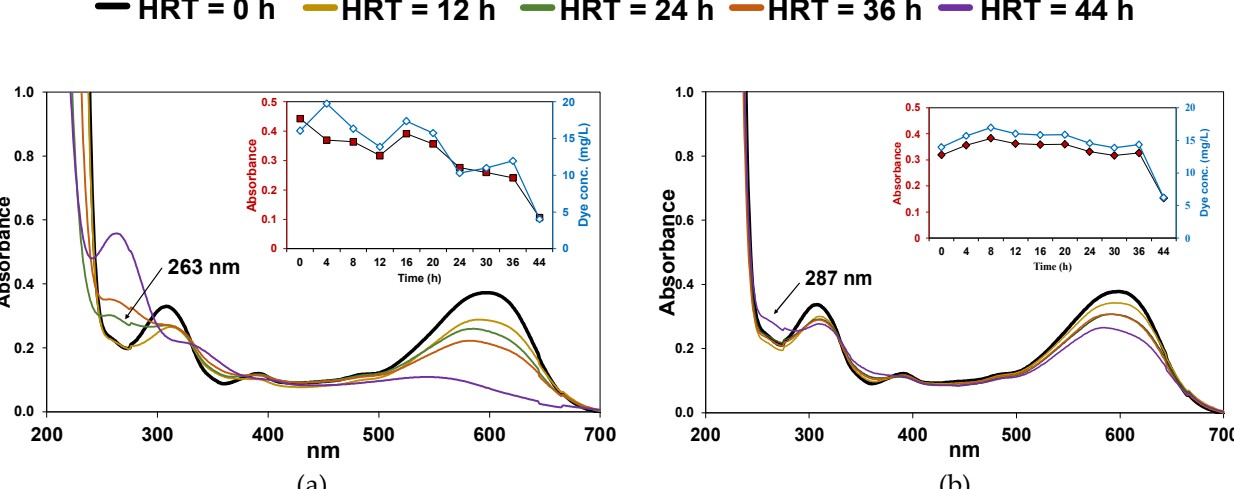

**Figure 6.** Time-overlaid UV-vis spectra for samples collected at 15 mg/L of RB-5 influent feeding in (**a**) reactor A and (**b**) reactor B; (inset) decrease in absorption at $\lambda_{max}$= 597 nm as a function of treatment period from run 3.

In conclusion, although RB-5 can be reduced through the HD process, gentle decolorization rates were observed, indicating inferior treatment efficiency due to the presence of $NO_3$–N and $NO_2$–N. However, a high rate of DRE was found after denitrification was completed, while increasing the initial concentration of dye led to a decrease in the degradation rate in this study.

### 3.4. Microbial Community Distribution under Various Operation Conditions

Initially, enhanced sludge from the HD reactor that had been operated for more than 400 days was added into both reactors A and B as initial sludge. Thereafter, these reactors were operated under various operating conditions for more than 60 days, as previously described [21]. These reactors were continuously operated in this study in run 1 (day zero).

The analysis of the phyla in the initial sludge showed that it contained 78.8, 10.3, 5.6, and 2.6% of Proteobacteria, Bacteroidetes, Firmicutes, and Planctomycetes, respectively. However, at the class level, there were slight differences in the microbial communities (Betaproteobacteria, 59.3%; Gammaproteobacteria, 12.8%; Cytophagia, 4.6%; and Alphaproteobacteria, 3.8%). The bacterial communities at the family and genus levels predominantly consisted of *Thauera* spp., belonging to the class, Betaproteobacteria, which was the most dominant bacterial community at the beginning stage of these reactors. After the analysis of the microbial communities in reactors A and B on day zero, analyses were again performed on days 20 (at the end of run 1) and 120 (at the end of run 5) so as to observe the effect of different $H_2$ flow rates, intermittent cycle times, and initial RB-5 dye concentrations on the microbial community structure in the sludge. At the phylum level, the results corresponding to both reactors were similar; the predominant phylum was close to Proteobacteria (32.4–87.2%) and Firmicutes (4.7–56.8%), whereas, at class level,

the microbial communities were as follows: Betaproteobacteria, 0.4–65.6%; Gammaproteobacteria, 4.1–79.8%; Alphaproteobacteria, 2.6–21.9%; and Clostridia,1.4–41.6%. It was observed that the $H_2$ flow rate and intermittent cycle times did not significantly affect the distribution of the microbial communities, suggesting that the distribution of Betaproteobacteria increased when the amount of RB-5 dye increased, whereas Gammaproteobacteria had a high distribution at low RB-5 concentrations. Additionally, Alphaproteobacteria and Clostridia showed high abundance in reactor B, especially on day zero. Previous studies suggested that the amount of $H_2$ and intermittent $H_2$ supply has an important effect on the distribution of the microbial communities in the HD systems, e.g., an increase in $H_2$ flow rate brought about an increase in the distribution of Betaproteobacteria, whereas, at low $H_2$ flow rates, Gammaproteobacteria and Alphaproteobacteria showed high abundance [42].

Further, the bacterial community was investigated in detail at family and genus levels (Figure 7). Specifically, the bacterial communities in both reactors were frequently identified as denitrification bacteria, autotrophic and heterotrophic denitrification bacteria, and anoxic and aerobic denitrifying bacteria [43,44]. These findings indicate that the differences in the gas supply conditions between anaerobic HD (reactor A) and combined aerobic/anaerobic HD (reactor B) did not have any significant effect on the distribution of the microbial communities except at day 0. Due to various operating conditions in each reactor, it might be affected by the different microbial communities in reactor A and B at the beginning stage. However, the distribution of microbial communities in both reactors was cleanly showed to be similar after days 20 to 100. The results suggested the convergence was induced by high concentrations of influent $NO_3$–N, initial dye concentrations, regardless of reactor types. The results presented an unclassified genus, Xanthomonadaceae, belonging to the classes Alphaproteobacteria and Rhodocyclaceae, as well as *Thauera* spp., belonging to the class Betaproteobacteria, which was the dominant bacterial community in both reactors. Moreover, some unclassified genera, Peptostreptococcaceae and Peptococcaceae, constituted the dominant bacterial communities on day 0 in reactor B. These bacteria represent a family of obligate anaerobic bacteria, some of which are facultative aerobic bacteria that play an important role in the removal of organic matter and are mostly found as dominant microorganisms during hydrolysis acidification [45,46].

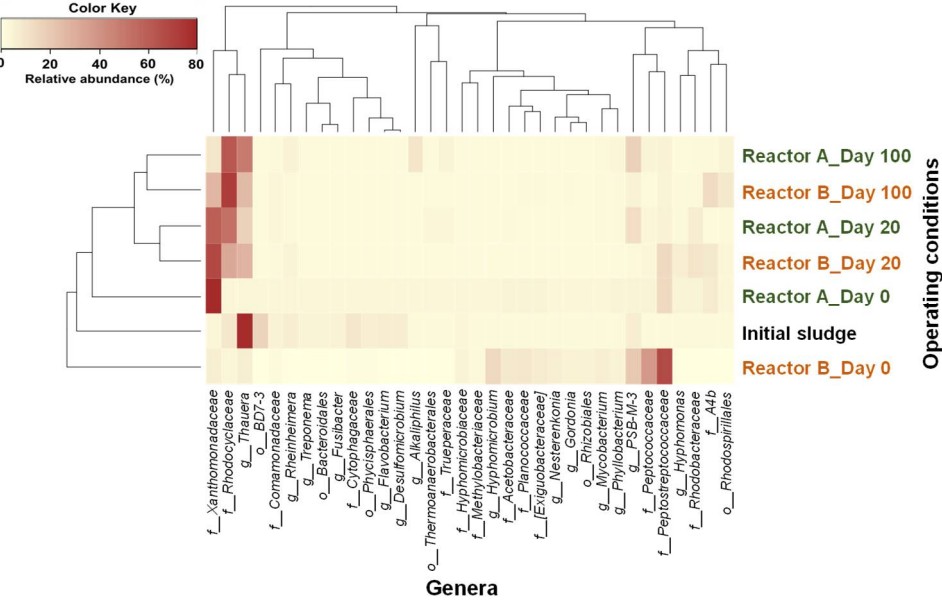

**Figure 7.** Heatmap of microbial community compositions (relative abundance in percentage) in sludge samples collected from reactors A and B under various operation times.

Given that the long aerobic conditions that characterized the beginning stage in reactor B might be unfavorable for Xanthomonadaceae, Rhodocyclaceae, and *Thauera* spp., they showed low relative abundances in this reactor as indicated in Table 3, confirming that the relative abundance of Xanthomonadaceae decreased as the RB-5 dye concentration increased. Specifically, the relative abundance of Xanthomonadaceae dropped from 37.2 to 4.3% in reactor A and from 39.0 to 16.1% in reactor B when the initial RB-5 concentration was increased from 10 to 60 mg/L. Moreover, it was observed that the relative abundance of Rhodocyclaceae increased as RB-5 concentration increased (from 32.9 to 36.6% and 18.0 to 46.1% in reactors A and B, respectively). Conversely, dye concentration did not affect the relative abundance of *Thauera* spp. Possibly, this is because these bacteria can adapt to contaminated environments and continuously grow following long-term exposure. However, it has been demonstrated that Xanthomonadaceae, Rhodocyclaceae, and *Thauera* spp. can grow under both anaerobic and combined aerobic/anaerobic conditions. The literature suggests that Xanthomonadaceae, which is considered a good candidate for the realization of autotrophic denitrification and is commonly found in high abundance in HD systems that involve the use of large amounts of salt and bicarbonate [47]. Reportedly, some of them participate in the biodegradation of phenol as well as in ammonia oxidation. Additionally, Rhodocyclaceae, which constitute a family of important denitrification bacterial species, play a key role in reducing $NO_3$ to $NO_2$ in HD systems. However, its relative abundance increases with increasing $H_2$ flow rate. Similarly, *Thauera* spp., which are frequently detected under autotrophic and heterotrophic denitrification conditions, are the main bacteria in HD systems. The results obtained suggested a decrease in the relative abundance of *Thauera* spp. at low $H_2$ flow rates. However, the relative abundance increased significantly when a high dosage of bicarbonate was used. In addition, most of the previously studied and approved *Thauera* spp. have been described as important aromatic compound degraders, and they show versatile aromatic compound-degrading capacity under denitrification conditions compared with aerobic conditions. Moreover, bacteria can use some intermediate metabolites after dye breaking as the sole carbon source in the decolorization process [48]

**Table 3.** Relative abundance of bacteria (%) under different operation conditions.

| Reactor/Sampling date (day) | Relative abundance of bacteria (%) | | | | | |
|---|---|---|---|---|---|---|
| | Reactor A. (Anaerobic) | | | Reactor B. (Combined Aerobic/Anaerobic) | | |
| | Day 0 | Day 20 | Day 100 | Day 0 | Day 20 | Day 100 |
| **Bacterial communities (Family and Genus)** | | | | | | |
| *Xanthomonadaceae*; Unclassified | 80.0 | 37.2 | 4.3 | 0.1 | 39.0 | 16.1 |
| *Rhodycyclaceae*; Unclassified | 0.1 | 32.9 | 36.6 | 1.4 | 18.0 | 46.1 |
| *Zoogloeaceae*; *Thauera* | 0.2 | 9.8 | 28.4 | 0.1 | 16.0 | 15.3 |
| *Peptococcaceae*; Unclassified | 0.4 | 1.0 | 1.3 | 15.2 | 1.1 | 0.1 |
| *Peptostreptococcaceae*; Unclassified | 10.0 | 2.1 | 2.0 | 26.4 | 7.3 | 11.3 |
| **Operation conditions** | | | | | | |
| $H_2$ flow rate (mL/min), Sparging cycle of air and $H_2$ (min/min) | 50 | 100 | 50 | 30/60 | 30/60 | 10/60 |
| Initial RB-5 dye conc. (mg/L) | - | 10 | 60 | - | 10 | 60 |

Therefore, the unclassified families, Xanthomonadaceae, Rhodocyclaceae, and *Thauera* spp. functioned as commensal bacteria that might be responsible for both anaerobic HD and combined aerobic/anaerobic HD, enhancing nitrogen removal performance and also reducing RB-5 dye color.

## 4. Conclusions

Two reactors, an anaerobic reactor (A) and a combined aerobic/anaerobic reactor (B), both with HD systems, i.e., autotrophic denitrification using $H_2$, were set up to evaluate the treatment process involving the removal of pollutants, especially $NO_3$, $NO_2$, and RB-5 dye, from textile industry wastewater. The effect of the important factors that affect NRE and DRE, namely, $H_2$ flow rate, the sparging cycle of air and $H_2$, and initial dye concentration, were investigated, and the results obtained showed a decrease in the concentrations of $NO_3$–N, $NO_2$–N, and RB-5 dye following the treatment of the textile wastewater in the HD system under both conditions. It was also observed that the $H_2$ flow rate plays a significant role in the realization of higher NREs and some color removal in the anaerobic HD reactor. Additionally, an increase in $H_2$ flow rate improved nitrogen removal as well as dye decolorization. This is because $H_2$ favors bacterial growth rate; thus, it controls denitrification efficiency. Similarly, it also acts as an electron source; hence, it helps to maintain the production of NADH, which is necessary for the generation of azo reductase that enables the transfer of electrons to an electron acceptor, such as a dye color via the carriers that are involved in the electron transport chains for the removal of dye color under bacterial degradation. However, $NO_3$–N and $NO_2$–N are electron acceptors that undergo faster oxidation–reduction reactions than the dye, which underwent decolorization to a limited extent in this study. Further, the results obtained confirmed that the dye color did not influence nitrogen removal; the biodegradation of RB-5 dyes only started after denitrification. Therefore, $NO_3$–N and $NO_2$–N concentrations significantly affected the decolorization rate. Furthermore, the sparging cycle of air and $H_2$ that was used to maintain air and $H_2$ gas flow under aerobic and anaerobic conditions influenced NRE and DRE in the combined aerobic/anaerobic reactor. Under anaerobic conditions, long HRTs resulted in higher treatment performance, given that in the HD system, oxygen can inhibit bacterial activity and can also adversely affect the enzymatic decolorization of azo dyes. Further, it was also observed that the level of dye color is an important factor; high RB-5 concentrations can cause poor nitrogen and color removal due to high toxicity. Fluctuations in NRE were observed when the reactors were operated for a short time; however, it remained at a higher level when the reactors were operated for a long time. It was also observed that high dosages had a small effect on NRE, especially the varying of the influent dye loading rate, and these high dosages required longer treatment times. Similarly, a low decolorization rate was observed when the initial dye concentration increased. Additionally, in terms of bacterial communities, unclassified genera belong to the families, Xanthomonadaceae, Rhodocyclaceae, and *Thauera* spp., representing the dominant bacterial communities in both reactors, played a significant role in enhancing the nitrogen and color removal rates.

In conclusion, this study demonstrated that HD systems could be applied to enhance textile wastewater treatment performance. It also showed that the factors, $H_2$ flow rate, sparging cycle of air and $H_2$, and initial dye concentration, are significant parameters that enhance nitrogen removal and decolorization efficiency in HD systems. Moreover, the findings of this study indicate that anaerobic and combined aerobic/anaerobic reactors have similar NRE and DRE; therefore, in a real situation involving the recovery of textile wastewater quality, a combined aerobic/anaerobic system is usually a better treatment process. This is because this wastewater mostly contains organic matter and some toxic chemicals that are usually removed in a combined system. Therefore, this study provides a plausible strategy for improving and developing advanced technologies that have low treatment costs and high treatment performance.

**Author Contributions:** Writing—original draft preparation, T.S.; writing—review and editing, K.S., S.R.; conceptualization, T.S., K.S., S.R., F.K.; supervision, F.K. All authors have read and agreed to the published version of the manuscript

**Funding:** This research received no external funding.

**Institutional Review Board Statement:** Not applicable.

**Data Availability Statement:** Data sharing not applicable.

**Acknowledgments:** The authors are thankful to the University of Yamanashi, Japan, for financial support and for the facilities provided through the Science and Technology Research Partnership for Sustainable Development (SATREPS) program of JST and JICA.

**Conflicts of Interest:** The authors declare no conflict of interest.

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
