# Peer review of "Factors Affecting the Simultaneous Removal of Nitrate and Reactive Black 5 Dye via Hydrogen-Based Denitrification"

_water, doi:10.3390/w13070922_

Round 1

Reviewer 1 Report

The present manuscript describes showed that the factors, H2 flow rate, sparging cycle of air and H2, and initial dye concentration, are significant parameters that enhance nitrogen removal and decolorization efficiency in HD systems and studied the corresponding relationship. The experimental test is detailed, and the result analysis is clear. In my opinion, this manuscript is worthy of publication as an article in Water. However, one issue should be considered before publication.

Problem:

“therefore, in a real situation involving the recovery of textile wastewater quality, a combined aerobic/anaerobic system is usually a better treatment process. This is because this wastewater mostly contains organic matter and some toxic chemicals that are usually removed in a combined system.” The proof of this conclusion requires references or experimental data.

Author Response

I already added the details and references at lines 64-67 and ref. NO. [11].

Reviewer 2 Report

In presented study, “Factors Affecting the Simultaneous Removal of Nitrate and Reactive Black 5 Dye via Hydrogen-based Denitrification”, authors investigated the effect of H2 flow rate, the sparging cycle of air and H2, and initial dye concentration on textile wastewater treatment of both aerobic and combined (aerobic/anaerobic) processes. I enjoyed reading of the article and appreciate authors effort for understanding and making better wastewater treatment system. There are one point I think authors should consider and that was about the interpretation of microbial community change. By improving the interpretation, the manuscript can bring more novel information for the field, in my opinion.

Detailed comments are following:

L101-110: This section of introduction is for explaining why studying both system is important (and this is the main point of current study). However, I am feeling that the current explanation presented is not fully convincing to understand why there should be combined treatment studied instead of further development of anaerobic treatment. Was this because authors wants to add aerobic treatment after anaerobic treatment? Or was this to explore better way to realize denitrification? It would be nicer if there is more direct comparison about the merits of combined treatment compared to anaerobic treatment or explain clearer what the authors are expecting.

L192-193: I guess authors used nanodrop via Quantus Fluorometer which offers both Qubit and nanodrop option. I think the description is inverted, please check.

L198: There was no description about the sequencing. Was it Miseq 250bp paired end sequencing? Or others?

L199-203: In post-sequencing data analysis, there is no description of normalization of data across the samples before the comparison. What kind of normalization methods applied before comparison? Rarefaction? Helinger transformation with abundance data? This should be described. Also, QIIME is a sort of pipeline, which is offering various way of processing the sequencing data. It will be better to provide more detailed description about how taxonomic classification was realized and how post sequencing quality trimming of sequences conducted. Without these information, it is hard to judge the interpretation of heatmap.

L405: RUN1 is capitalized, while in other part of manuscript it was not.

L415: it is confusing that the text says Day 0, Day 20, and Day 100, but figure says Day 0, Day 20, and Day 120. This should be improved.

L426: “Previous studied” -> Previous studies

L435-438: Considering the Figure 7 and Table 3, for me the data say that there was a huge difference between the microbial community from reactor A and B at Day 0. This can reflect some difference caused by system of reactor A and B as authors try to emphasize. However, considering all other data of current study, the data say there was a convergence of microbial community to be similar to each other  at Day 120 even though Day 0 community clearly showed extreme difference. This convergence was already clearly observed in Day 20 samples (which didn’t show significant difference to Day 120), and this implies the observed convergence is irrelevant with all those changes among the runs (1 to 5). Instead, the data suggests, the convergence was induced by high concentration of influent NO3-N or dye, regardless of reactor types. This is very striking result for me as normally biologist would expect clear divergence of microbial community between anaerobic and aerobic condition. The discussed difference in bacterial taxa between reaction A and B, in the line of shifting from Day 0 to 120, is hard to be directly compared by numbers as the sequencing was not having replication. However, I do think the observed dominant bacterial taxa are interesting as they are the result of this induced environment of contamination even overcoming general selective pressure expected between anaerobic and aerobic microbial community. I think authors should put more effort to interpret this result and develop further discussion.

Author Response

Comments and Suggestions for Authors

Reviewer 2:

In presented study, “Factors Affecting the Simultaneous Removal of Nitrate and Reactive Black 5 Dye via Hydrogen-based Denitrification”, authors investigated the effect of H2 flow rate, the sparging cycle of air and H2, and initial dye concentration on textile wastewater treatment of both aerobic and combined (aerobic/anaerobic) processes. I enjoyed reading of the article and appreciate authors effort for understanding and making better wastewater treatment system. There are one point I think authors should consider and that was about the interpretation of microbial community change. By improving the interpretation, the manuscript can bring more novel information for the field, in my opinion.

Detailed comments are following:

L101-110: This section of introduction is for explaining why studying both system is important (and this is the main point of current study). However, I am feeling that the current explanation presented is not fully convincing to understand why there should be combined treatment studied instead of further development of anaerobic treatment. Was this because authors wants to add aerobic treatment after anaerobic treatment? Or was this to explore better way to realize denitrification? It would be nicer if there is more direct comparison about the merits of combined treatment compared to anaerobic treatment or explain clearer what the authors are expecting.

 A: I added more seasons that why the previous research select a combined treatment system to compared the treatment performance with anaerobic condition.

“Additionally, previous studied attempted to develop the HD process and evaluated its performance in relation to the simultaneous treatment of various pollutants in textile wastewater, especially NO3-, NO2-, and dye [20]. Their results showed the possibility of the simultaneous reduction of nitrogen and a small amount of dye in both anaerobic and combined aerobic/anaerobic reactors. In an anaerobic reactor that was operated by continuous feeding with H2 gas, they observed the complete removal of NO3- and NO2- as well as a decrease in the color intensity of the RB-5 dye. Similarly, a combined aerobic/anaerobic reactor that is commonly used to reduce organic/inorganic pollutant contents in textile wastewater was set-up by feeding with air and H2 gas to evaluate the treatment efficiency of the HD system coupled with aerobic conditions mainly to find out other ways to reduce cost of H2 gas supply. In real wastewater treatment plants, most aerobic bacteria play a significant role in the removal of organic pollutants as well as some toxic chemicals from wastewater.”

L192-193: I guess authors used nanodrop via Quantus Fluorometer which offers both Qubit and nanodrop option. I think the description is inverted, please check.

A: Sludge samples, with a wet weight of approximately 0.12 g, were collected for total DNA extraction using a FastDNA® SPIN Kit for Soil analysis (MP-Biomedicals, USA).DNA concentration was determined by nanodrop using a QuantusTM Fluorometer (Promega Corp., USA), and DNA samples were used for next-generation sequencing (NGS) analysis that was carried out by a commercial service in Japan (FASMAC Co., Ltd.).

L198: There was no description about the sequencing. Was it Miseq 250bp paired end sequencing? Or others?

 A: The Miseq platform was used to obtain the amplified metagenomic sequences.

“In this details of this method, I did not known clearly because we sent the samples to company, and also they did not give more details about this method.”

L199-203: In post-sequencing data analysis, there is no description of normalization of data across the samples before the comparison. What kind of normalization methods applied before comparison? Rarefaction? Hellinger transformation with abundance data? This should be described.

Also, QIIME is a sort of pipeline, which is offering various way of processing the sequencing data. It will be better to provide more detailed description about how taxonomic classification was realized and how post sequencing quality trimming of sequences conducted. Without these information, it is hard to judge the interpretation of heatmap.

 A: The amplified metagenomic sequences were obtained using the Miseq platform. The raw sequence data was then taxonomically classified using QIIME software version 1.9.0 to obtain the operational taxonomic units (OTUs), which were then clustered at 97% similarity, and the bacterial relative abundance.

“Following the results that I got from the company, they were sent as the final results in unit of then abundance data follow phylum, class, order, family and gene. Then, I transfer the results to percent of relative abundance (%), and remove some results that lower than 1%, then, make a heatmap.

L405: RUN1 is capitalized, while in other part of manuscript it was not.

A: edited

L415: it is confusing that the text says Day 0, Day 20, and Day 100, but figure says Day 0, Day 20, and Day 120. This should be improved.

A: edited

L426: “Previous studied” -> Previous studies

A: edited

L435-438: Considering the Figure 7 and Table 3, for me the data say that there was a huge difference between the microbial community from reactor A and B at Day 0.

This can reflect some difference caused by system of reactor A and B as authors try to emphasize.

However, considering all other data of current study, the data say there was a convergence of microbial community to be similar to each other at Day 120 even though Day 0 community clearly showed extreme difference.

This convergence was already clearly observed in Day 20 samples (which didn’t show significant difference to Day 120), and this implies the observed convergence is irrelevant with all those changes among the runs (1 to 5).

 Instead, the data suggests, the convergence was induced by high concentration of influent NO3-N or dye, regardless of reactor types.

This is very striking result for me as normally biologist would expect clear divergence of microbial community between anaerobic and aerobic condition.

The discussed difference in bacterial taxa between reaction A and B, in the line of shifting from Day 0 to 120, is hard to be directly compared by numbers as the sequencing was not having replication.

However, I do think the observed dominant bacterial taxa are interesting as they are the result of this induced environment of contamination even overcoming general selective pressure expected between anaerobic and aerobic microbial community.

I think authors should put more effort to interpret this result and develop further discussion.

 A: These findings indicate that the differences in the gas supply conditions between anaerobic HD (reactor A) and combined aerobic/anaerobic HD (reactor B) did not have any significant effect on the distribution of the microbial communities except at Day 0. Due to various operating conditions in each reactor, it might be effected on the different microbial community in reactor A and B at beginning stage. However, the distribution of microbial communities in both reactors were cleanly showed to be similar after day 20 to 100. The results suggested the convergence was induced by high concentrations of influent NO3-N, dye concentrations, regardless of reactor types.
